# N2 sleep promotes the occurrence of 'aha' moments in a perceptual insight task

**Anika T. Löwe**[1,2,3☯*], **Marit Petzka**[1,2,3☯], **Maria M. Tzegka**[2,3], **Nicolas W. Schuck**[1,2,3‡]

**1** Institute of Psychology, Universität Hamburg, Hamburg, Germany, **2** Max Planck Research Group NeuroCode, Max Planck Institute for Human Development, Berlin, Germany, **3** Max Planck UCL Centre for Computational Psychiatry and Ageing Research, Berlin, Germany

‡ Senior author.
☯ These authors contributed equally to this work.
* loewe@mpib-berlin.mpg.de

**Data availability statement:** Behavioral data is publicly available here on figshare: https://doi.org/10.6084/m9.figshare.28806383.

## Abstract

Humans sometimes have an insight that leads to a sudden and drastic performance improvement on the task they are working on. The precise origins of such insights are unknown. Some evidence has shown that sleep facilitates insights, while other work has not found such a relationship. One recent suggestion that could explain this mixed evidence is that different sleep stages have differential effects on insight. In addition, computational work has suggested that neural variability and regularisation play a role in increasing the likelihood of insight. To investigate the link between insight and different sleep stages as well as regularisation, we conducted a preregistered study in which N=90 participants performed a perceptual insight task before and after a 20 minute daytime nap. Sleep EEG data showed that N2 sleep, but not N1 sleep, increases the likelihood of insight after a nap, suggesting a specific role of deeper sleep. Exploratory analyses of EEG power spectra showed that spectral slopes could predict insight beyond sleep stages, which is broadly in line with theoretical suggestions of a link between insight and regularisation. In combination, our findings point towards a role of N2 sleep and aperiodic, but not oscillatory, neural activity for insight.

## Introduction

Having an insight, or aha-moment, is a unique learning phenomenon that has attracted researchers' interest for a century [1]. The cognitive and neural mechanisms that underlie insight are still debated [2,3], and have for instance been described as a restructuring of existing task representations [4–6]. On a behavioural level, insight is often characterised by three features: an abrupt, non-linear increase in task performance [7,8]; a variable delay before the insight occurs 'spontaneously' [6]; and selective occurrence in only some, but not all participants [9,10].

An important milestone along the path to understanding insight will be to define the factors that facilitate its occurrence. One such potential factor is sleep, which is linked to memory consolidation [11] and restructuring of memories [12], suggesting that it could be a facilitating factor for the incubation of insight. The evidence that sleep supports insight, however,

EEG data is also available on Figshare: https://doi.org/10.6084/m9.figshare.28805639. All analysis code is publicly available here https://github.com/maritp/sleepsight and linked to the following DOI: https://doi.org/10.5281/zenodo.15299273. The preregistration can be found here: https://osf.io/z5rxg.

**Funding:** ATL is supported by the International Max Planck Research School on Computational Methods in Psychiatry and Ageing Research (IMPRS COMP2PSYCH, www.mps-ucl-centre.mpg.de). NWS was funded by the Federal Government of Germany and the State of Hamburg as part of the Excellence Initiative, a Starting Grant from the European Union (ERC-StG-REPLAY-852669), and an Independent Max Planck Research Group grant awarded by the Max Planck Society (M.TN.A.BILD0004). We acknowledge financial support from the Open Access Publication Fund of Universität Hamburg. The funders had no role in study design, data collection and analysis, decision to publish, or preparation of the manuscript.

**Competing interests:** The authors have declared that no competing interests exist.

is inconclusive. Work by Wagner et al. [13] suggests a beneficial effect of a full night's sleep on insight, finding that more than twice as many subjects gained insight into a hidden task rule after sleep, compared to wakefulness. Another study reported similar findings after a daytime nap [14]. Other investigations, in contrast, did not find any benefits of sleep for insight, or reported no difference between sleep and awake rest [15–17].

One possibility to explain divergent findings is that particular sleep stages affect insight in different ways. Lacaux et al. [14] investigated this question by letting participants have a daytime nap in between sessions of a mathematical insight task, where discovering a hidden rule allowed to solve the task much more efficiently. In this case, a beneficial effect of sleep on insight was associated exclusively with sleep stage 1 (N1) [14], which led to a 83% probability to discover the hidden rule, compared to 30% in participants who stayed awake and 14% in those how reached deeper N2 sleep. Another possibility is that different cognitive tasks benefit differently from sleep. Lerner et al. [18], for instance, have argued that sleep is particularly important for extracting hidden task regularities, in line with work that has found the strongest sleep effects in tasks that require extracting statistical rules. This would suggest that insight tasks, which mainly rely on learning implicit associative rules (such as the PSSST we tested in this study), benefit more from sleep.

Given the diverging findings on the impact of sleep on insight, we conducted a preregistered daytime nap intervention study based on procedures by Lacaux et al. [14], but used a different task (pregregistration link: https://osf.io/z5rxg/resources). We first aimed to replicate the above mentioned finding that N1 sleep compared to wakefulness after task exposure would lead to a higher number of insight moments about a hidden strategy during the post-nap behavioural measurement, while N2 sleep would lead to a reduced number of insight moments. A second major interest was to understand which features of the sleep-EEG signal best predict insight. Past work has focused on power in individual frequency bands [14]. However, our own computational work [10] has suggested that a combination of regularisation and noise had beneficial effects for insight. Regularisation is a technique commonly used in machine learning that involves shrinking weights towards zero, in an attempt to avoid overfitting [19]. Our interest in regularisation is based on the observation that homeostatic plasticity processes during sleep can lead to global shrinkage of synaptic strengths [20], in a manner that is broadly reminiscent of regularisation [21]. This shrinkage could be critical for insights in several ways. Our own modelling work has shown that regularisation causes an initial suppression of irrelevant features that leads to the delayed and abrupt nature of knowledge development during insight [10]. Additionally, we speculate that a global downscaling of synaptic weights during sleep could lead to a "cleaner slate" that might facilitate insight post nap. While a direct mapping between noise or regularisation in neural networks and electrophysiological signals is unknown, the concepts of noise [22] and regularisation (as in synaptic downscaling, [23]) have been indirectly linked to aperiodic activity [24]. Additionally, aperiodic activity has been shown to decrease with an increase in sleep depth [23,25,26]. Hence, we also asked whether aperiodic activity of the EEG signal might have additional effects on insight, over and above the hypothesised relations to sleep stages.

Instead of the Number Reduction Task (NRT) employed by Lacaux et al. [14], we employed the Perceptual Spontaneous Strategy Switch Task (PSSST) that also features a hidden task regularity, and which our previous work has shown to invoke insight-based spontaneous strategy switches [9,10,27]. Similarly to the NRT, participants initially learned a functional, but suboptimal, strategy, which was replaced by some participants with a more optimal solution through an insight [9,27–29].

We note that while our task has the benefit to allow for tracking insight on a trial basis, it also differs from other tests in which participants are asked to actively search for a novel

problem solution (e.g. Remotes Associates Tasks [30] or Compounds Remotes Associates Tasks [31]).

## Materials and methods

### Participants

Participants between eighteen and 35 years of age were recruited via internal mailing lists as well as the research participation platform Castellum. Participation in the study was contingent on not having any learning difficulty nor colour blindness. Further, participants needed to report a normal sleep-wake cycle and no history of sleep disorders. Participants were excluded if they switched to the colour strategy immediately after the correlation onset, before the nap. All participants gave written informed consent prior to beginning the experiment. The study protocol was approved by the local ethics committee of the Max Planck Institute for Human Development (approval number i2022-15) and adhered to the Declaration of Helsinki. Participants received 56€ for completing the entire experimental procedure.

Data inclusion was contingent on participants' showing learning of the stimulus classification. As in our previous study with the PSSST [10], we probed their accuracy on the three easiest, least noisiest coherence levels in the last block of the uncorrelated task phase. 30 subjects did not reach an accuracy level of at least 80% in those trials and were thus excluded from further analyses. Fifteen subjects were excluded, because the gained insight before the nap and further 7 subjects were excluded due to insufficient EEG data quality. The final sample included in all analyses thus contains 68 datasets.

### Behavioural task

**Perceptual Spontaneous Strategy Switch Task (PSSST).**  We employed the PSSST used in our previous work [9,10,27] that requires a binary choice about circular arrays of moving dots [32], but adapted the motion coherence levels slightly. Dots were characterised by two features, (1) a motion direction (four possible orthogonal directions: NW, NE, SW, SE) and (2) a colour (orange or purple). The noise level of the motion feature was varied in 5 steps (5%, 23%, 41%, 59% or 76% coherent motion), making motion judgement relatively harder or easier. Colour difficulty was constant, thus consistently allowing easy identification of the stimulus colour. The condition with most noise (5% coherence) occurred slightly more frequently than the other conditions (30 trial per 100, vs 10, 20, 20, 20 for the other conditions).

The task was coded in JavaScript and made use of the jsPsych 6.1.0 plugins. Stimuli were presented on a 24 inch screen with a resolution of 1920 x 1200 pixel and a refresh rate of 59 Hz. On every trial, participants were presented a cloud of 200 moving dots with a radius of 7 pixels each. In order to avoid tracking of individual dots, dots had a lifetime of 10 frames before they were replaced. Within the circle shape of 400 pixel width, a single dot moved 6 pixel lengths in a given frame. Each dot was either designated to be coherent or incoherent and remained so throughout all frames in the display, whereby each incoherent dot followed a randomly designated alternative direction of motion.

The trial duration was 2000 ms and a response could be made at any point during that time window. After a response had been made via one of the two button presses, the white fixation cross at the centre of the stimulus turned into a binary feedback symbol (happy or sad smiley) that was displayed until the end of the trial. An inter trial interval (ITI) of either 400, 600, 800 or 1000 ms was randomly selected. If no response was made, a "TOO SLOW" feedback was displayed for 300 ms before being replaced by the fixation cross for the remaining time of the ITI.

**RDK task design.** For the first 350 trials, the *motion phase*, the correct binary choice was only related to stimulus motion (two directions each on a diagonal were mapped onto one choice), while the colour changed randomly from trial to trial. For the binary choice, participants were given two response keys, "X" and "M". The NW and SE motion directions corresponded to a left key press ("X"), while NE and SW corresponded to a right key press ("M"). Participants received trial-wise binary feedback (correct or incorrect), and therefore could learn which choice they had to make in response to which motion direction.

We did not specifically instruct participants to pay attention to the motion direction. Instead, we instructed them to learn how to classify the moving dot clouds using the two response keys, so that they would maximise their number of correct choices. To ensure that participants pick up on the motion relevance and the correct stimulus-response mapping, motion coherence was set to be at 100% in the first block (100 trials), meaning that all dots moved towards one coherent direction. In the second task block, we introduced the lowest, and therefore easiest, three levels of motion noise (41%, 59% and 76% coherent motion), before starting to use all five noise levels in block 3. Since choices during this phase should become solely dependent on motion, they should be affected by the level of motion noise.

After the *motion phase*, in the *motion and colour phase*, the colour feature became predictive of the correct choice in addition to the motion feature. This means that each response key, and thus motion direction diagonal, was consistently paired with one colour, and that colour was fully predictive of the required choice. Orange henceforth corresponded to a correct "X" key press and a NW/SE motion direction, while purple was predictive of a correct "M" key press and NE/SW motion direction. This change in feature relevance was not announced to participants, and the task continued for another 550 trials as before - the only change being the predictiveness of colour.

Before the last task block we asked participants whether they 1) noticed the colour rule in the experiment, 2) how long it took until they noticed it, 3) whether they used the colour feature to make their choices and 4) to replicate the mapping between stimulus colour and motion directions. We then instructed them about the correct colour mapping and asked them to rely on colour for the last task block. This served as a proof that subjects were in principle able to do the task based on the colour feature and to show that, based on this easier task strategy, accuracy should be near ceiling for all participants in the last instructed block.

**Psychomotor Vigilance Task (PVT).** During the PVT, a white fixation cross was presented in the middle of the screen. After a delay (jittered with 4000±2000 ms), the fixation cross changed its colour to red. The change in colour prompted participants to press the space key as fast as possible. On key press, participants received feedback about their reaction time for 2.5 s. Overall, the PVT comprised 25 trials, corresponding to approximately 3 min. For results of the PVT see Fig E in S1 Text.

## Experimental procedure

For all participants, the experimental procedure started at the same time (1 pm) to rule out potential time of day confounds. The experimental procedure consisted of 3 parts: (1) a first behavioural session of about 25 minutes, including the PVT and 400 trials of the RDK task, followed by (2) a nap of 20 minutes and (3) a second behavioural session of about 30 minutes, including the PVT and 500 more trials of the RDK task. Note that the original PSSST version does not have a delay.

(1) The experimental procedure began with the Pittsburgh Sleep Quality Index (PSQI) questionnaire. Participants then first completed the PVT and concluded with the first part of the RDK task of which the last 50 trials contained the hidden, easier strategy.

(2) Subsequently, participants were given time to rest and nap for 20 minutes. The EEG cabin was a completely dark and noise shielded room without sensory stimulation. During the nap break, participants were positioned in a semi-reclined position on an armchair with their legs resting on a foot piece, holding a light plastic cup in one hand. With the onset of N2-sleep this cup likely falls, waking participants up (see [14]). EEG recordings were exclusively recorded during this period and were used to identify different sleep stages. To increase the probability that people would fall asleep during the nap, sleep in the night before the experiment was reduced by 30% and participants were additionally asked to refrain from consuming caffeine prior to the session. All participants started the session at the same time of day at 1 pm.

(3) After the nap, participants resumed the behavioural testing and first performed a second PVT, followed by 500 more trials of the RDK task.

## Object

We used the same object as Lacaux and colleagues [14] for this experiment: a light (55 g) plastic drinking cup with a height of 14.5 cm and a 5.5 cm diameter. A babyphone filming space below the hand of the participant next to the armchair was used to get accurate time stamps of the object drop should the cup fall out of the participant's hand due to muscle tonus relaxation.

## Modelling of insight-like switches

To investigate insight based strategy adaptations, we modelled participants' data using individually fitted sigmoid functions (for details see [10]).

$$y = \frac{y_{max} - y_{min}}{1 + e^{-m(t-t_s)}} + y_{min}$$

The criterion defined in order to assess whether a subject switched to the colour strategy, is the accuracy in the highest noise level (5% coherence) in the last task block before the colour rule was explicitly instructed. Insight subjects are classified as those participants whose performance on those trials was above 85%. The individual insight moments $t_s$ were derived from the individually fitted sigmoid functions.

## EEG recordings

During the nap period, EEG and electrooculography (EOG) data were recorded using a Brain Products 64-channel EEG system with a sampling rate of 1000 Hz. All electrodes were referenced online to A2 (right mastoid) and AFz was used as the ground electrode. Two external electrodes (biploar reference and ground electrode on the forehead) were placed on the chin to record muscle activity (electromyography, EMG). Impedances were kept below 20 kΩ.

## Sleep scoring

EEG and EOG data were re-referenced offline to linked mastoids and band pass filtered between 0.3 and 35 Hz (high pass filter: 0.3 Hz, two-pass butterworth filter, 3rd order; low pass filter: 35 Hz, two-pass butterworth filter, 5th order). EMG data were high pass filtered at 5 Hz (two-pass butterworth filter, 3rd order). Lastly, all data were down-sampled to 200 Hz.

To identify different sleep stages, sleep was scored according to the guidelines from the American Academy of Sleep Medicine (AASM, [33]) based on EEG (O2, O1, Pz, Cz, C3, C4,

F3 and F4), EOG and EMG data. Participants without any N1 or N2 period were assigned to the wake group. Participants who had at least 1 epoch (30 s) of N1 and no signs of N2 (sleep spindles and/or K-complexes) were assigned to the N1 group. Participants with signs of N2 (sleep spindles and/or K-complexes) were assigned to the N2 group. For the AASM scoring, 30 s epochs were used. Scoring was done by two scorers (ATL and MP), blind to the experimental condition. Additionally, we validated the scoring by a convolutional neural network trained on external polysomnography data (U-Sleep, [34], correlation with manual scoring: $r(66) = 0.82$, $p<0.001$).

In addition to sleep stages, Lacaux et al. [14] reported a modulation of insight by alpha and delta power across the whole nap period. To test for an additional modulation of insight by power of different frequency ranges, we used a data driven approach across the frequency spectrum of 1–20 Hz (see section Spectral slope analysis).

## EEG data analysis

EEG analyses were conducted using the FieldTrip toolbox [35] and custom scripts written in MATLAB. Independent component analysis (ICA) was applied to remove eye movement artifacts from the data. For that, data were re-referenced offline to linked mastoids, filtered (two-pass butterworth filter: high-pass: 1Hz, low-pass: 100Hz, bandstop: 48-52Hz) and down-sampled (200 Hz). Bad channels were removed and coarse artifacts were discarded based on outliers regarding amplitude and variance (implemented in *ftrejectvisual*). ICA was applied to identify components reflecting eye movements (saved together with the unmixing matrix). The raw data were then pre-processed again since previous pre-processing was optimised for ICA. Data were re-referenced to linked mastoids, filtered (two-pass butterworth filter: high-pass: 0.1Hz, low-pass: 48Hz) and down-sampled (200Hz). Bad channels were removed and the previously obtained unmixing matrix was applied to the data, components reflecting eye movements were removed and data were demeaned. Finally, bad channels were interpolated (spherical spline interpolation) and artifacts were visually identified.

**Spectral slope analysis.** To obtain estimates of aperiodic activity, the spectral slope parameter $x$ (reflecting the slope of the power spectrum) was used. Data of the whole 20 minute nap period were segmented into 6 second epochs with an overlap of 50%. For these segments, power spectra were obtained by applying a Hanning window and transforming data from time to frequency domain using Fast Fourier Transformation. Power spectra were calculated for 1–45 Hz with a frequency resolution of 0.2Hz. The FOOOF algorithm [36] was then applied to obtain the spectral slope. Aperiodic activity $a$ is defined by:

$$a = 10^b * \frac{1}{(k + f^{\frac{1}{x}})}$$

where $b$ is the y intercept, $k$ is the knee parameter and $x$ is the slope parameter. The knee parameter was set to 0 since we did not expect the aperiodic activity to change across the frequency range. Lower frequencies (< 1 Hz) often show a plateau in the power spectrum resulting in the need to fit a knee [37]. To avoid this, the lower limit of the frequency range was set to 1 Hz. All other settings for the fit (including the peaks for the periodic component) were kept at their default values (maximum number of peaks = 3, minimum height of a peak = 3 dB, limits of peak width = 0.5–12 Hz).

## Statistical analyses

Fisher's exact tests were used in the analysis of contingency tables. All tests were two-tailed with a significance level of less than 0.05. All computations were performed using R version 4.3.1. For comparisons of spectral slopes between Wake, N1 and N2 groups or between participants with vs. without insight across all channels, a cluster-based permutation test was used (F-statistics for comparison between Wake, N1 and N2: 1000 permutation, alpha = 0.05 , clusteralpha = 0.05; t-statistics for comparison between Insight vs. No insight: 1000 permutation, alpha = 0.025 , clusteralpha = 0.05). For post-hoc comparisons, t-tests were applied. For model comparisons, we used the following logistic regression models for each EEG channel:

$Model_{baseline}$: Insight $\sim$ 1 + sleep stage

$Model_1$: Insight $\sim$ 1 + sleep stage + slope

$Model_2$: Insight $\sim$ 1 + slope

AIC scores and likelihood ratio tests were used to assess the best model fit in a hierarchical manner, while adjusted *McFaddensR$^2$* scores are reported to indicate total model fit.

Bayes Factors were analysed in R using the BFpack library using fitted GLMs for the respective hypotheses as inputs. The respective hypotheses were included as constraints.

## Results

To study the effect of different sleep stages on insight, 90 participants performed a previously developed perceptual insight task, [9], before and after a 20-minute nap break. Subjects were presented with a stimulus consisting of dots that were (1) either orange or purple (colour feature) and (2) moved in one of four possible orthogonal directions (motion feature, see Fig 1A). Dot motion had a varying degree of noise across trials (5%, 23%, 41%, 59% or 76% coherent motion), making motion judgement relatively harder or easier on different trials. Participants were instructed to learn the correct button for each stimulus from trial-wise binary feedback (see Fig 1A and 1B). The main task consisted of 9 blocks of 100 trials each in which participants had to press one of two buttons in response to the shown stimulus, and observe the feedback afterwards.

In the first three task blocks, only stimulus motion correlated with the correct response, such that the correct button was deterministically mapped onto the directions of the dots (two directions for each response). However, starting in the middle of block 4, stimulus colour began predicting the correct button as well (i.e. the colour was paired with the two directions that predicted the same response button, see Fig 2A). After block 4, participants were given an opportunity to nap for 20 minutes in a reclining arm chair. We monitored brain activity and sleep during this phase using a 64-channel electroencephalography (EEG). Participants then completed 5 more blocks of the task, during which colour continued to predict the correct response in addition to motion (Fig 2A). Additional details about the task can be found in the Methods section.

The subtle, unannounced change in task structure after 3.5 blocks provided a hidden opportunity to improve the decision strategy that could be discovered through insight. Insight was spontaneous in the sense that participants were not instructed about the hidden rule and did not need to switch their strategy to perform the task correctly. Only after a participant incidentally discovered the hidden rule did it become clear that using the colour could make the task easier.

We tracked insight on a trial-by-trial basis by monitoring rapid performance increases on high-noise (i.e. low motion coherence) trials, on which accuracy prior to the onset of colour predictiveness was at only 56% (vs. 92% in low noise trials; how accuracy depended on the noise level is shown in Fig 1D). Performance in high noise trials was stable before the

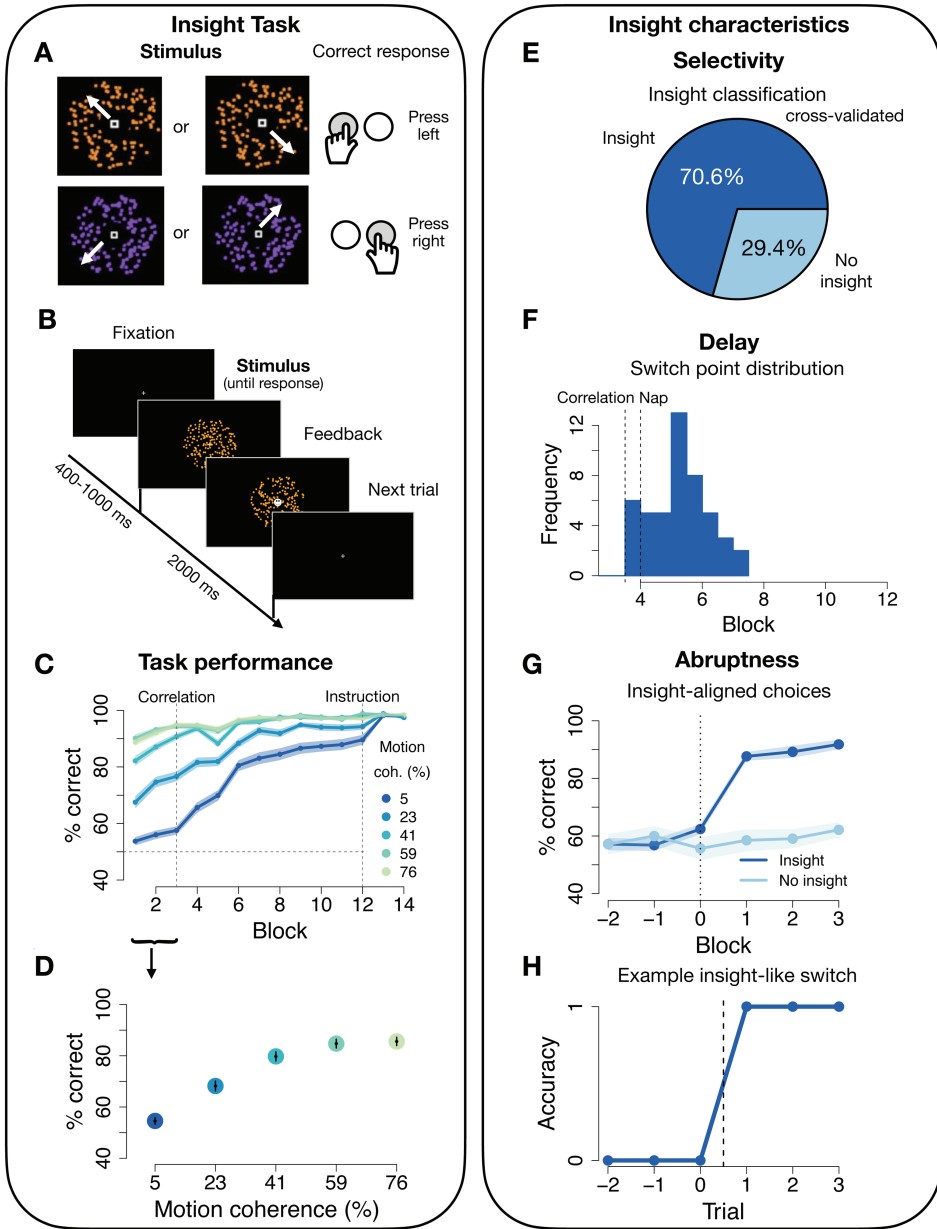

**Fig 1. Performance on the PSSST insight task and behavioural insight characteristics.** A: Stimuli and stimulus-response mapping of the PSSST. Dot clouds were either coloured in orange or purple and moved to one of the four directions (NW, NE, SE, SW) with varying coherence. A left response key, "X", corresponded to the NW/SE motion directions, while a right response key "M" corresponded to NE/SW directions. **B:** Trial structure: a fixation cue is shown for a duration that is shuffled between 400, 600, 800 and 1000 ms. The random dot cloud stimulus is displayed for 2000 ms. A response can be made during these entire 2000 ms, but a central feedback cue will replace the fixation cue immediately after a response. **C:** Accuracy (% correct) over the course of the experiment for all motion coherence levels. The first dashed vertical line marks the onset of the colour correlation, the second dashed vertical line the instruction about colour predictiveness. Blocks shown are halved task blocks (50 trials each). N = 90, error shadows signify standard error of the mean (SEM). **D:** Accuracy (% correct) during the motion phase increases with increasing motion coherence. N = 90, error bars signify SEM. **E:** 70.6% of subjects (48/68) were classified as insight subjects based on non-linear increases in performance on the lowest motion coherence level (5%). **F:** Distribution of switch points. The first dashed vertical line marks onset of the colour correlation, the second dashed vertical line the nap period. Blocks shown are halved task blocks (50 trials each). **G:** Switch point-aligned accuracy on the lowest motion coherence level for insight (48/68) and no-insight (20/68) subjects. Blocks shown are halved task blocks (50 trials each). Error shadow signifies SEM. **H:** Trial-wise switch-aligned binary responses on lowest motion coherence level for an example insight subject. The underlying data for this figure can be found at https://doi.org/10.6084/m9.figshare.28806383.

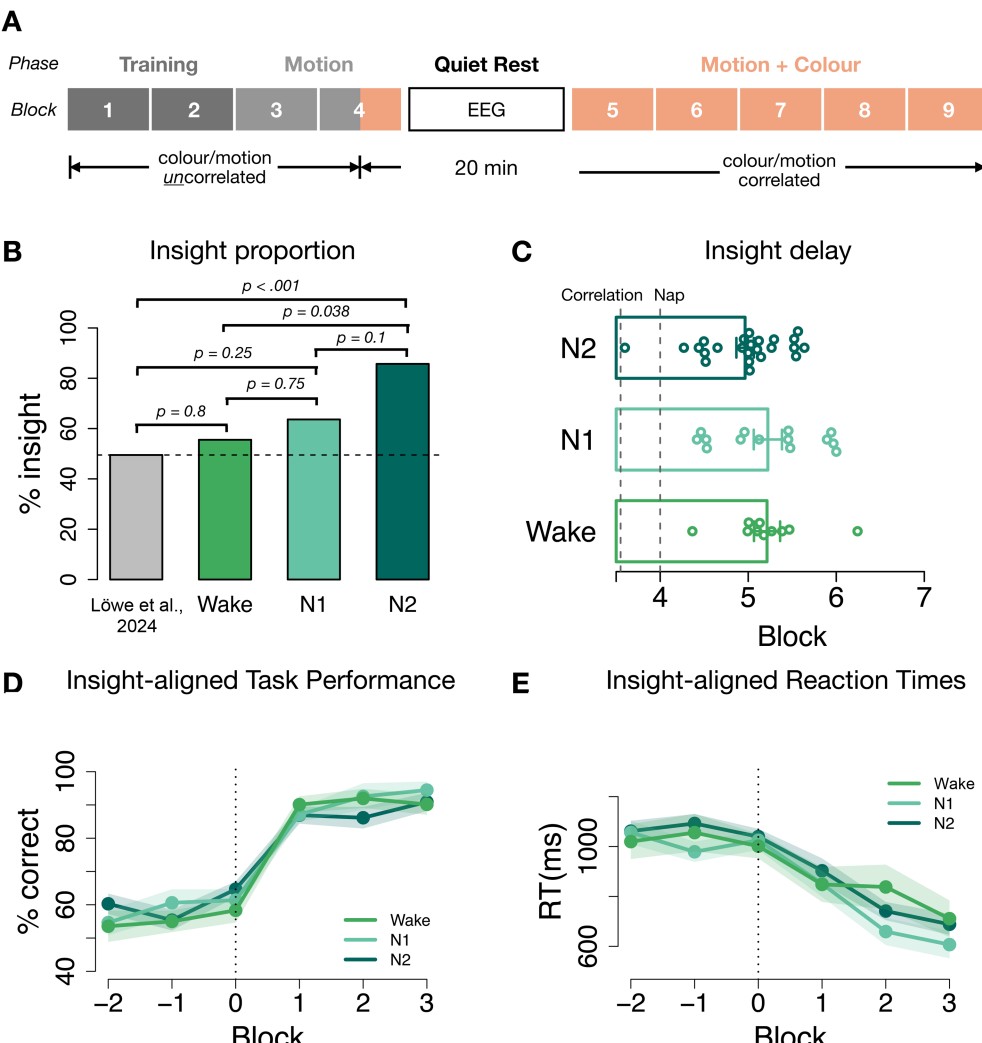

**Fig 2. PSSST task structure and insight across sleep groups.** A: Task structure of the PSSST: each block consisted of 100 trials. A first training block contained only 100% motion coherence trials to familiarise subjects with the S-R mapping. The remaining training block contained only high coherence (41%, 59%,76%) trials. In the motion phase, colour changed randomly and was not predictive and all motion coherence levels were included. Colour started to be predictive of correct choices and correlate with motion directions as well as correct response buttons in the second half of the 4th block to expose subjects to the hidden rule before the nap. Participants were then given 20 minutes to nap while EEG was recorded. Before the very last block 9, which served as sanity check, participants were instructed to use colour. **B:** Insight proportion among the different sleep groups. The insight ratio was significantly higher for the N2 sleep group (85.7%) than for the Wake group (55.5%). The N1 sleep group ratio (63.6%) did not differ significantly from the other two groups. The insight baseline ratio of 49.5% was derived from our previous work using the same task without any nap or other delay period. **C:** Distribution of switch points for the different sleep groups. One beeswarm point is one insight participant. Barplots show the mean, error bars signify SEM. **D:** Switch point-aligned accuracy and **E:** reaction times on the lowest motion coherence level for insight subjects of the respective sleep groups. Blocks shown are halved task blocks (50 trials each). Error shadow signifies SEM. The underlying data for this figure can be found at https://doi.org/10.6084/m9.figshare.28806383.

change in task structure (paired t-test first half of block 3 vs. first half of block 4: 55% vs. 58%, $t(157.8) = -1.51$, $p = 0.13$, $d = 0.23$, Fig 1C), indicating that improvements do not arise simply due to training. A sudden change towards high accuracy on high noise trials can therefore be interpreted as indicative of insight about the colour-based strategy [9,10,27].

## 20 Minutes of rest increase insight

Fifteen subjects had an insight before the nap and were therefore excluded from analysis. In another 7 cases EEG data quality prevented sleep classification, resulting in a total of 68 subjects for post nap data analysis. 70.6% (48/68) of participants showed abrupt, non-linear performance improvements after the nap and were thus classified as "insight participants" (Fig 1E). Notably, this percentage is substantially higher than a baseline of 49.5% (49/99) insight that we observed in our previous study with closely related experimental procedures, but without any delay period ($p$ = .007, Fisher's exact test, see Fig 2B below; N = 99, data from [10]). By the first half of block 8, insight participants had significantly higher average accuracy across all trial types ($M$ = 98.2 $\pm$ 0.3% vs $M$ = 86.4 $\pm$ 0.9%, $t(22.86)$ = 12.28, $p$<.001, $d$ = 4.26), and lower reaction times ($M$ = 526.6$\pm$14 vs $M$ = 767.4$\pm$30.4, $t(27.4)$ = −7.19, $p$<.001, $d$ = 2.2), as expected. Hence, the 20 minute nap period significantly improved insight. Insight showed all three characteristics we observed in previous work: First, insight was selective, i.e. occurred only in some, but not all, participants (see above). Second, the timing of individual strategy switch points differed substantially across participants, indicating the highly variable delay known as impasse in the insight literature (block in which switch occurred: $M$ = 5.1 $\pm$ 2.6, range 3.6–6.2, Fig 1F; analyses based on logistic function fits, see Methods). Third, if participants had an insight, their accuracy increased very abruptly within a short time window, i.e. time-locking performance to their individual switch point indicated an average 25% performance jump within merely 15 trials ($M$ = 62.4 $\pm$ 16.9% vs $M$ = 87.6 $\pm$ 15.1%, $t(92.8)$ = −11.16, $p$<.001, Fig 1G), which often reflected performance changes within a single trial only (Fig 1H).

## No evidence for N1 but for N2 sleep promoting insight

We followed the procedure of Lacaux et al. [14] and divided participants into three groups based on their vigilance state during rest. Sleep was manually scored according to the guidelines from the American Academy of Sleep Medicine [33] based on 30 s EEG (O2, O1, Pz, Cz, C3, C4, F3 and F4), EOG and EMG epochs. Using these criteria, participants were categorised as having had either no sleep, N1 sleep, or N2 sleep. This analysis showed that during the 20 minute nap period 28 participants reached N2 sleep, 22 reached only N1 sleep, and 18 subjects remained awake (Please note that one subject spent 1 min in N3 sleep, but was nevertheless included in the N2 group; for sleep characteristics see Table A in S1 Text). Within the N2 group, 85.7% (24/28) gained insight into the hidden strategy, while only 63.6% (14/22) of participants in the N1 group and 55.5% (10/18) of the Wake group gained an insight in our task (Fig 2B). Since manual sleep stage scoring depends on subjective classification, we validated the manual sleep stage scoring with a convolutional neural network trained on external polysomnography data (U-Sleep, [34]). This categorisation correlated highly with manual scoring, $r(66)$ = 0.82, $p$<0.001), and results reported here can be replicated qualitatively using this alternative approach (see S1 Text). To further argue for the reported results being independent of the sleep stage scoring technique, we split participants based on their subjective sleep reports. Again, the reported results can be qualitatively replicated (see Fig B in S1 Text), although subjective reports did not match objective sleep staging closely (see S1 Text).

Based on the paper by Lacaux et al. [14], our main preregistered hypothesis proposed that N1 sleep would lead to an increased number of insight compared to the Wake and N2 sleep groups, respectively. We further hypothesised that N2 sleep would lead to decreased insight compared to N1. We find no support for either the first or second hypothesis (Fisher's exact test N1 vs. Wake: $p$ = 0.75; N1 vs. N2: $p$ = 0.1). To explain the above reported heightened incidence of insight after the nap generally, we explored whether N2 sleep was the main driver

of insight. Interestingly, we observed a significantly higher number of insight after N2 sleep compared to Wake (Fisher's exact test, $p = 0.038$, Fig 2B). In line with these analyses, a generalised linear model (GLM) with sleep stage as a predictor of insight fits the data better than a model with just an intercept (AIC 82.5 vs. 84.4). As expected, post-hoc tests also showed a significant N2 sleep coefficient in this model ($p = 0.03$), while N1 sleep and Wake remained non-significant (Wake: $p = 0.64$, N1: $p = 0.6$). Investigating Bayes Factors supports this finding and shows strong evidence for an effect of N2 > N1 (BF = 24.71) as well as N2 > Wake (BF = 8.19), while there is no substantial evidence for our preregistered hypotheses of N1 > W (BF = 1.19) and N1 > N2 (BF = 0.04). We thus find no evidence that N1 sleep promotes insight as reported by Lacaux et al. [14]. Instead, in our data N2 sleep showed a significant association with insight frequency. Please note that the results do not qualitatively change when participants with very short episodes of N2 sleep (< 2min) are excluded (excluded participants = 4; insight likelihood of $N2_{adjusted}$ = 87.5% [21 of 24 participants with insight], Fisher's exact test $N2_{adjusted}$ vs. N1, $p = 0.09$; Fisher's exact test $N2_{adjusted}$ vs. W, $p = 0.03$).

The increased occurrence of insight in the N2 group had no major associations with overall performance after the nap. Accuracy on the lowest motion coherence trials only trended to be better in N2 compared to Wake participants (t-test block 5-12, N2 vs. Wake: $M = 85 \pm 3\%$ vs $M = 76 \pm 2.9\%$, $t(14) = 2.06$, $p = 0.06$, $d = 1.03$, N2 vs. N1: $M = 85 \pm 3\%$ vs $M = 81 \pm 2\%$, $t(12.1) = 1.06$, $p = 0.31$, $d = 0.53$, Fig BA in S1 Text).

No effects on the corresponding reaction times could be found (N2 vs. Wake: $M = 757.6 \pm 48$ms vs $M = 809 \pm 35$ms, $t(12.8) = -0.86$, $p = 0.4$, $d = 0.43$, N2 vs. N1: $M = 757.6 \pm 48$ms vs $M = 787.8 \pm 45$ms, $t(13.9) = -0.46$, $p = 0.66$, $d = 0.23$, Fig BB in S1 Text). Finally, there was no significant difference in the vigilance between Wake, N1 and N2 (assessed via reaction times in a psychomotor vigilance task, PVT, before and after the nap. linear model: $rt_{log} \sim 1 + $ sleep stage + time point; for all $\beta$: $-0.045 < \beta < 0.031$, all $p > 0.2$; Details see Methods and Fig E in S1 Text). Thus, sleep seemed to increase insight frequency, but not alter overall performance characteristics.

To explore more directly whether the characteristics of insight differed between sleep groups, we next focused on the individually determined time points of insight, and participants' performance thereafter. We investigated differences in delay using the individually defined switch points in high noise trials (Fig 1G and 1F; details see Methods), and found no significant differences across groups ($M_{N2} = 4.96 \pm 0.1\%$; $M_{N1} = 5.22 \pm 0.16\%$; $M_{Wake} = 5.21 \pm 0.15\%$, see Fig 2C; all $t$s <1.39, $p$s >.18). The switch point distributions also did not differ between groups (Kolmogorov-Smirnov test: N1–Wake: $D = 0.33$, $p = 0.47$, N1–N2: $D = 0.29$, $p = 0.36$, N2–Wake: $D = 0.33$, $p = 0.36$). Accuracy of insight subjects after their switch did not differ between sleep groups either ($M_{N2} = 90.9 \pm 0.3\%$; $M_{N1} = 94.5 \pm 0.3\%$; $M_{Wake} = 90.2 \pm 0.3\%$, see Fig 2D; all $t$s <1.06, $p$s >.3). Finally, we also found no group differences between reaction times after the insight ($M_{N2} = 688.4 \pm 42$; $M_{N1} = 607.1 \pm 54.7$; $M_{Wake} = 711 \pm 73$, see Fig 2E; all $t$s <−0.27, $p$s >.25). Thus, while N2 sleep increased the prevalence of insight, it does not seem to affect its characteristics, i.e. abruptness, selectivity and delay.

## No evidence for oscillatory activity predicting insight

Additionally to sleep stages, Lacaux et al. [14] found an association between insight and alpha and delta power. We pre-registered a data-driven analysis approach (including frequencies from 1–20 Hz) to test for a modulation of insight by power. To this end, we contrasted spectral slope corrected power spectra (FOOOF algorithm [36], 6 s epochs, 1–20 Hz, 0.2 Hz frequency resolution, 50% overlap) between Wake, N1 and N2. Power spectra were calculated as described in the Methods section (Spectral slope analysis).

As expected, oscillatory power in the frequency range of 6–16 Hz significantly differed across all channels between Wake, N1 and N2 (cluster-based permutation test, F-statistics, $p_{cluster}$ = 0.005). Post hoc cluster-based permutation tests revealed a positive and negative cluster in the alpha (5.8–11.3 Hz) and sleep spindle frequency range (11.5–15.2 Hz), respectively (post-hoc cluster-based permutation test, t-statistics, Wake > N1: negative cluster, $p_{cluster}$ = 0.02, 10.5–14 Hz; Wake > N2: positive cluster, $p_{cluster}$ = 0.07, 6–9 Hz; negative cluster, $p_{cluster}$ = 0.05, 11.5–15.2 Hz; N1 > N2: positive cluster, $p_{cluster}$ = 0.007, 5.8–12.3 Hz). Since we also wanted to test the a priori hypothesis of delta power predicting insight (see pre-registration), we additionally extracted power in the delta frequency range (1–4 Hz). Neither averaged power in the alpha nor in the spindle cluster nor in the delta frequency range explained insight beyond sleep stages (AIC for model containing only sleep stages = 82.5; AIC for model with sleep stages + alpha power at channel C4 = 84.5; AIC for model with sleep stages + spindle power at channel C4 = 84.1, Fig DB in S1 Text; AIC for model with sleep stages + delta power at channel C4 = 84.4). In line with the spectral slope analyses, we also removed sleep stages from both models. Removing sleep stages from both models resulted in a worse model fit (AIC for model with spindle power at channel C4 = 85.9; AIC for model with spindle power at channel C4 = 86.3, Fig DC in S1 Text; AIC for model with delta power at channel C4 = 85.6). A complementary pattern emerges when directly contrasting participants with and without insight across the whole frequency range. No significant differences were observed, neither in the alpha, spindle, delta or any other frequency range (cluster-based permutation test, $p_{cluster}$ = 0.31).

Together, these results suggest that oscillatory activity does not explain insight, neither alone nor in combination with sleep stages.

## Aperiodic neural activity predicts insight

Above, we performed pre-registered analyses investigating sleep stages and their impact on insight. They revealed that N2 sleep in particular is associated with insight. In a next step, we follow up on these findings with exploratory analyses investigating a potential association between insight and aperiodic activity. Our previous work on neural networks [10] suggests that noise as well as regularisation facilitate sudden and abrupt performance changes characterising insight. Although the precise mapping of these parameters in neural networks onto electrophysiological markers is unclear, noise [22] and regularisation (as in synaptic down-scaling, [23]) have both been associated with aperiodic activity. Additionally, aperiodic activity has been shown to decrease along the sleep cycle, translating into a steeper spectral slope with deeper sleep [23,25,26]. This led us to ask whether aperiodic activity during the nap period relates to insight, over and above the effects of sleep stages. We quantified aperiodic neural activity during the entire 20 min nap period by the spectral slope of the power spectrum in log-log space (FOOOF algorithm by range 1–45 Hz, 0.2 Hz frequency resolution, 6 s epochs with 50% overlap [36]). Analyses that use only aperiodic neural activity during the deepest sleep stage (e.g., N1 sleep stage for N1 group) revealed qualitatively similarly results to those reported below and are shown in Fig F in S1 Text. We verified that spectral slopes differ between the Wake, N1 and N2 groups, as expected [23,25,26]. This showed a global association (across all channels) between the spectral slope and sleep stages ($p_{cluster}$ = 0.003) such that the spectral slope was the steepest in the N2 group and the flattest in the Wake group (post-hoc t-tests, for channel F4: Wake vs. N1: $M_{Wake}$ = −1.37 ± 0.08 vs. $M_{N1}$ = −1.47 ± 0.04, $t(26.9)$ = 1.72, $p$ = 0.25, $d$ = 0.39, N1 vs. N2: $M_{N1}$ = −1.47 ± 0.04 vs. $M_{N2}$ − 1.79 ± 0.06, $t(47.3)$ = 4.33, $p<0.001$, $d$ = 1.18; for channel C4: Wake vs. N1: $M_{Wake}$ = −1.31 ± 0.09 vs. $M_{N1}$ = −1.50 ± 0.05,

$t(26.8) = 2.00$, $p = 0.06$, $d = 0.67$, N1 vs. N2: $M_{N1} = -1.50 \pm 0.05$ vs. $M_{N2} - 1.76 \pm 0.06$, $t(47.8) = 3.49$, $p = 0.001$, $d = 0.96$, Fig 3A).

Our main question was whether the spectral slope relates to insight beyond the association between sleep stages and insight reported above. Given the substantial association between sleep stages and spectral slope, we used a nested model comparison approach and tested a baseline model containing only sleep stage as a predictor for insight against a full model containing sleep stage and spectral slope. While the sleep stage only model performed better than an intercept-only null model (for channel F4: AIC null model: 84.4, AIC sleep stage model: 82.5; likelihood ratio test between null and sleep stage model: $X^2(2) = 5.85$, $p = 0.05$, for sleep stage$_{N1vsWake}$, $\beta = 0.34$, $p = 0.60$, for sleep stage$_{N2vsWake}$, $\beta = 1.57$, $p = 0.03$), we also found that spectral slope over fronto-central areas further improved insight prediction compared to the sleep stage only model (for channel F4: AIC: 82.5 vs. 81.0, Fig 3B, likelihood ratio test between both models: $X^2(1) = 3.56$, $p = 0.06$), with a steeper spectral slope relating to a higher insight likelihood (e.g., channel F4: $\beta = -1.98$, $p = 0.07$). Interestingly, comparing this full model (with

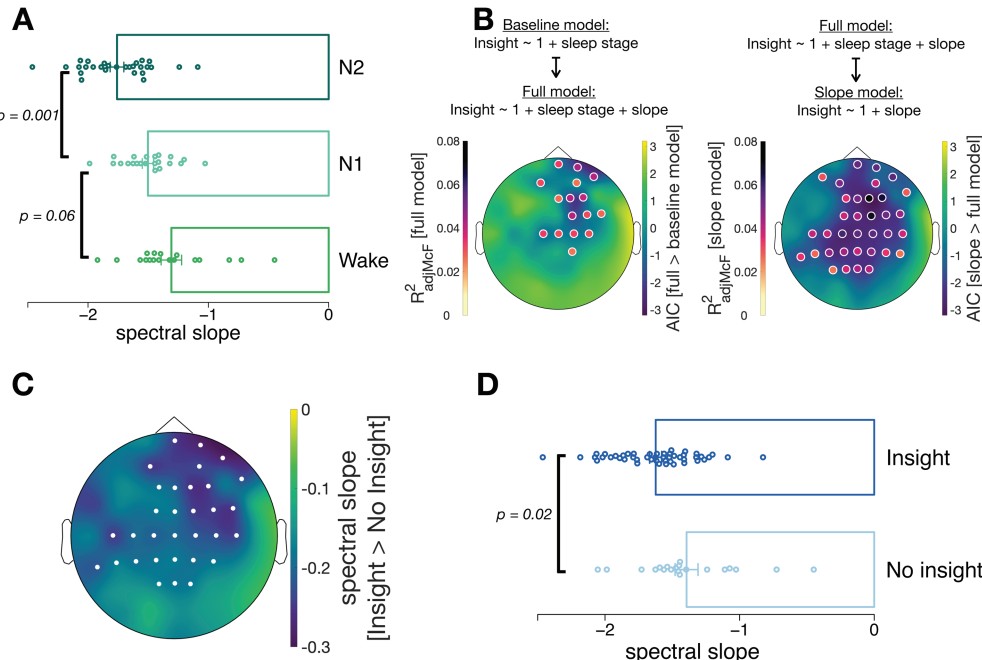

**Fig 3. Spectral slope analysis.** A: The spectral slope significantly decreased from Wake to N1 to N2, as expected. For the corresponding topoplot see Fig C in S1 Text. **B:** Topographies of model comparison results testing the full model, including sleep stage and spectral slope, vs. a baseline model, including just sleep stage (left) and the slope model, including just the slope, vs. the full model, including sleep stage and spectral slope (right). Shown are channel-wise model fit improvements (see colour scale on the right side) obtained by including the spectral slope (left, AIC differences with negative values indicate a better fit of the full model) or removing sleep stage (right, AIC differences with negative values indicate a better fit of the slope model). For channels with AIC differences <0 and a significant model prediction of the full (left) or slope model (right), *McFaddensR*$^2$ adjusted for model complexity is shown as a colour-coded dot (see colour legend left). See Table B in S1 Text for model information of frontal, central, parietal and occipital channels. **C:** The spectral slope was significantly steeper (i.e., more negative) for participants with insight vs. participants without insight, over fronto-central areas. All channels that are part of the significant cluster are highlighted in white. **D:** The comparison of the spectral slope between participants with vs. without an insight for channel C4 (part of the significant cluster in C). Repeating all analyses with aperiodic neural activity calculated only during the deepest sleep stage revealed comparable results, see Fig F in S1 Text. The underlying data for this figure can be found at https://doi.org/10.6084/m9.figshare.28805639.

both sleep stage and spectral slope as predictors) with the more parsimonious model containing only the spectral slope showed that the spectral slope alone is the best predictor for insight, yielding the best of all considered models (e.g., channel F4: AIC: 81.0 vs. 78.2, Fig 3B, for slope, $\beta = -2.53$, $p = 0.01$, likelihood ratio test that the more complex model is better: $X^2(2) = 1.25$, $p = 0.54$). As anticipated based on these results, contrasting participants with versus without insight also indicated clear differences in spectral slope ($p_{cluster} = 0.01$, Fig 3C; for channel F4: Insight vs. No Insight: $M_{Insight} = -1.65 \pm 0.04$ vs. $M_{NoInsight} = -1.40 \pm 0.08$, $t(30.85) = -2.69$, $p = 0.01$, $d = 0.77$; for channel C4: Insight vs. No Insight: $M_{Insight} = -1.62 \pm 0.04$ vs. $M_{NoInsight} = -1.39 \pm 0.09$, $t(30.05) = -2.39$, $p = 0.02$, $d = 0.69$, Fig 3D).

Investigation of oscillatory activity, in contrast, did not reveal any correlation with insight. Although oscillatory activity changed across sleep stages, and Lacaux et al. [14] reported links between alpha and delta power and insight, we did not find such associations in our data (see above for an overview of the analyses).

In conclusion, variations in aperiodic activity during a nap period predict whether participants will gain insight, with steeper spectral slopes, particularly over fronto-central areas, linked to higher insight likelihood. Aperiodic activity explained insight beyond sleep stages and was even sufficient to explain insight to a similar extent than aperiodic activity together with sleep stages, with the latter suggesting that aperiodic activity captures all information obtained by sleep stages. Further, the association between aperiodic activity and insight is stronger than previously described links between sleep stages or oscillatory power and insight.

## Discussion

We investigated the effect of sleep on insight. Our preregistered study set out to conceptually replicate findings of Lacaux et al. [14], who reported that effects of sleep on insight were driven entirely by N1 sleep. While we did find a general increase in insight following the nap, the insight ratio of N1 subjects did not differ from subjects of the Wake group, thus providing no support for the hypothesis that N1 sleep fosters insight, contrary to [14]. Instead, we found a beneficial effect of N2 sleep on post-nap insight likelihood, suggesting a need for deeper sleep for insight. We note that although the combination of awake and sleep group differences in this study and previous observations from studies without any delay strongly suggest a beneficial effect of sleep rather than delay, we do not present a randomized manipulation of sleep, awake rest, and no rest.

An exploratory analysis showed that the 1/f slope of the power spectrum did explain additional variance in insight likelihood above and beyond sleep stages. In contrast, neither power in the alpha nor in the spindle frequency range could predict insight. Hence, aperiodic but not oscillatory neural activity emerged as an additional factor that promotes insight. The 1/f slope has been linked to consciousness and sleep depth, where a steeper slope signifies less consciousness under anaesthesia. From wakefulness, to N1, N2 and N3 sleep, i.e., when sleep becomes deeper, the spectral slope becomes steeper [25,38–41]. Since the 1/f slope is a continuous measurement and tracks different brain states during sleep on a short timescale [42], it potentially offers a more fine grained measure of sleep depth. Hence, the fact that the spectral slope predicts insight beyond sleep stages alone suggests that deeper sleep is needed for insight.

This begs the question what the insight promoting processes during deeper sleep are. Our previous computational work [10] pointed towards a role of regularisation and noise for the formation of insight. Regularisation is a model simplification process that is commonly used in machine learning to improve performance and avoid overfitting [19]. In the context of neural networks, regularisation shrinks or eliminates in particular weak weights between

neurons. On a molecular level, this renormalisation has been linked to synaptic downscaling during sleep [24], a process that maintains a synaptic firing homeostasis by adjusting synaptic weights based on activity [20]. It has been specifically argued that by regulating synaptic strength depending on the neurons' firing rates during wake, this scaling process can aid stable energy requirements and may avoid memory interference [20]. Proponents of the synaptic homeostasis hypothesis [43–45] have linked regularisation to synaptic downscaling [21], since both processes reduce weights and support generalisation. The selective, activity-dependent renormalisation of synapses through regularisation during sleep might thus be an ideal candidate for gist extraction of relevant information and therefore contribute to insight-like learning phenomena.

By pruning synaptic connections with low activity, overall excitability is renormalised during sleep [46–48]. Computational work correlated this excitation-inhibition (E/I) balance with the spectral slope of aperiodic EEG activity [24] where a reduced E/I balance is reflected in a steeper spectral slope. Beyond just being a fine grained measure of sleep depth, the 1/f slope might thus reflect regularisation, which potentially plays an important role in generating insight.

It should be noted, however, that to date it is unclear if synaptic downscaling occurs during NREM sleep. Some evidence has linked E/I balance adjustments to REM sleep [23], but also to slow-wave sleep [49], while further evidence for synaptic downscaling during NREM sleep has remained indirect [50,51]. While our nap intervention was not sufficiently long for participants to reach N3 sleep, our results are in line with several findings suggesting that deeper sleep more generally plays a special role in the origins of insight. Increased alpha band activity during slow wave sleep (SWS), for instance, has been found to be predictive of post sleep insight, potentially reflecting representational restructuring happening during deeper sleep [52]. Another study found a SWS-specific effect in the beta frequency range, but interpreted particularly the oscillatory 10 Hz patterns during SWS to imply neocortical read-out of implicitly learned information stored in the hippocampus [53]. While we do not directly observe SWS in our study, and individual frequency bands such as alpha did not predict insight, the evidence seems to broadly suggest that deeper forms of sleep yield bigger benefits for insights. How this can be reconciled with findings from Lacaux et al. [14] and others has yet to be determined. Future work should further investigate the role of sleep beyond NREM and include a full night of sleep.

What amount of regularisation is beneficial for insight is also uncertain. While our previous work [10] has suggested that a certain amount of regularisation in neural networks leads to abrupt learning dynamics that characterise insight, either too little or too much regularisation caused the network to behave less insight-like. In the present study we only found a one directional relation, where deeper sleep and thus possibly more regularisation predicted insight. A speculative explanation for this might be that downscaling during N2 sleep of the nap led to a sort of reset of the previously learned synaptic weights which led participants to have a 'clean slate' after the nap, enabling them to restart the task with a fresh mind and discover the hidden rule more easily.

Lastly, why our findings diverge from what was reported by Lacaux et al. [14] is unclear. A major difference between our studies is that we used the Perceptual Spontaneous Strategy Switch Task [9,10,27], while they used the NRT. The PSSST has crucial analogies in task structure to the NRT. Both tasks measure 'intrinsic' insight where the hidden rule as a potential for strategy improvement is never mentioned to participants, and both tasks can be solved in principle even if the hidden rule is not discovered, by using the initially learned rule. Besides the fact that our rule was much simpler, there are two major differences between these two insight tasks: first, the initial rule was learned via feedback in the PSSST, while it

was instructed in the NRT; second, in our study the hidden rule became possible only after 350 trials, while for the NRT it is present from the start. This could imply potentially different learning mechanisms that could be affected differently by the respective sleep stages. Further, Lacaux et al. [14] use occipital electrodes for oscillatory analyses, but our spectral slope results find an effect of aperiodic activity predicting insight on fronto-central electrodes (Fig 3C).

While such differences do not allow inferences about the original finding, conceptual replications are important for validating broader scientific implications. How theoretical constructs such as insight are mapped onto specific tasks needs to be carefully evaluated, if one seeks to test the theoretical construct of interest.

Beyond sleep stages, different cognitive tasks might benefit from sleep in different ways – and potentially explain divergent findings regarding the relationship of sleep and insight. For instance, sleep has been found to aid in statistical learning [54] and particularly in the extraction of hidden task regularities [18]. Besides the associative learning of hidden structure in the PSSST and NRT, participants also need to overcome the initially learned strategy, which could block learning of the associations required for the insight [55,56]. Both downscaling of redundant information in terms of weak synaptic connections, as well as global downscaling of all synaptic weights to a speculative 'clean slate' could potentially account for the post sleep insight. In favour of the clean slate hypothesis would be that subjects often take a while after waking up to discover the hidden rule [13,14], i.e. they don't wake up with the insight solution.

It is important to note that regularisation only reflects one of several possible mechanisms that foster insights and are not mutually exclusive. As discussed above, regularisation can lead to a "cleaner slate" after a nap, akin to a process that relaxes previously established constraints, and can lead to gist extraction. Another mechanism that could foster insight is replay, the reactivation of sequential experience during sleep or wake [57], which can be used to recombine previously separate experiences and thereby promote novel inferences [58,59]. Replay on the other hand, might be beneficial for generating new connections, such as would be required for insight puzzles. It would also be expected that based on this mechanism, subjects should know the solution upon waking and restarting the task.

Further studies on the relationship between sleep and insight should therefore continue to evaluate different tasks, for instance one that is neither mathematical nor perceptual. Additionally, future work could also investigate the effect of a full night of sleep, rather than brief naps.

To conclude, the present study presents evidence of N2 sleep increasing insight likelihood, with the EEG spectral slope predicting insight beyond sleep stages. An exciting avenue for future studies will be to investigate the mapping between on-task EEG activity during insight moments to EEG activity during sleep and further examine potential relationships between the EEG spectral slope and regularisation in neural networks.

## Supporting information

**S1 Text. Supporting text.** Supplementary information file including additional analyses on self-reported sleep stages and sleep stages using U-Sleep [34]. This file includes Fig A (Accuracy and reaction times on the lowest motion coherence level for subjects of the respective sleep groups), Fig B (Insight proportion, switch point distribution, accuracy and reaction times among the reported sleep groups), Fig C (F-values of the comparison of the spectral slope between Wake, N1 and N2), Fig D (Oscillatory activity and topographies of model comparison results), Fig E (PVT results), Fig F (Results of quantifying aperiodic neural activity for the deepest sleep stage), Table A (Average sleep durations), Table B (Information about

the full model and the slope model across frontal, central, parietal and occipital area). Figure legends see inside S1 Text.
(ZIP)

## Author contributions

**Conceptualization:** Anika T. Löwe, Marit Petzka, Nicolas W. Schuck.

**Data curation:** Anika T. Löwe, Marit Petzka.

**Formal analysis:** Anika T. Löwe, Marit Petzka, Nicolas W. Schuck.

**Funding acquisition:** Nicolas W. Schuck.

**Investigation:** Anika T. Löwe, Marit Petzka, Maria M. Tzegka.

**Methodology:** Anika T. Löwe, Marit Petzka, Nicolas W. Schuck.

**Project administration:** Anika T. Löwe.

**Resources:** Nicolas W. Schuck.

**Supervision:** Nicolas W. Schuck.

**Validation:** Anika T. Löwe, Marit Petzka.

**Visualization:** Anika T. Löwe, Marit Petzka.

**Writing – original draft:** Anika T. Löwe, Marit Petzka, Nicolas W. Schuck.

**Writing – review & editing:** Anika T. Löwe, Marit Petzka, Nicolas W. Schuck.

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
