## [Editor Report · Decision Letter 0]

5 Sep 2024

Dear Dr Schuck, 

Thank you for submitting your manuscript entitled "N2 Sleep Inspires Insight" for consideration as a Research Article by PLOS Biology.

Your manuscript has now been evaluated by the PLOS Biology editorial staff as well as by an academic editor with relevant expertise and I am writing to let you know that we would like to send your submission out for external peer review.

Once your full submission is complete, your paper will undergo a series of checks in preparation for peer review. After your manuscript has passed the checks it will be sent out for review. To provide the metadata for your submission, please Login to Editorial Manager (https://www.editorialmanager.com/pbiology) within two working days, i.e. by Sep 07 2024 11:59PM.

Kind regards,

Suzanne

Suzanne De Bruijn, PhD, 

Associate Editor

PLOS Biology

sbruijn@plos.org

---

## [Decision Letter · Decision Letter 1]

6 Nov 2024

Dear Dr Schuck,

Thank you for your patience while your manuscript "N2 Sleep Inspires Insight" was peer-reviewed at PLOS Biology. It has now been evaluated by the PLOS Biology editors, an Academic Editor with relevant expertise, and by several independent reviewers. 

In light of the reviews, which you will find at the end of this email, we would like to invite you to revise the work to thoroughly address the reviewers' reports. 

As you will see below, the reviewers thought this an interesting study, but do have some concerns, and requests for clarifications, which we would like you to address.

Given the extent of revision needed, we cannot make a decision about publication until we have seen the revised manuscript and your response to the reviewers' comments. Your revised manuscript is likely to be sent for further evaluation by all or a subset of the reviewers.

**IMPORTANT - SUBMITTING YOUR REVISION**

*Re-submission Checklist*

*Published Peer Review*

*PLOS Data Policy*

*Blot and Gel Data Policy*

Sincerely,

Suzanne

Suzanne De Bruijn, PhD 

Associate Editor

PLOS Biology

sbruijn@plos.org

REVIEWS:

Reviewer #1: This is a highly interesting paper investigating the role of sleep in insight. It is very well written, the methods are sound and I truly enjoyed the read! What I find particularly intriguing about this work is that it suggests a novel mechanism by which sleep may facilitate insight. While previous work attributed sleep's role in insight to replay/reactivation related processes, this paper strongly suggests that - on the contrary - mechanisms of synaptic renormalization during sleep may facilitate later insight into a hidden structure governing a task. This is in keeping with suggestions from the extant literature on insight and AHA moments and will probably spark much discussion in the field.

I only have a few minor comments that might strengthen the manuscript and improve readability for a larger audience: 

1. In addition to sleep stage, may also the kind of task/task structure matter in how sleep contributes to insight? Could it be that task where renormalization or a relaxation of constraints is key benefit most strongly from sleep? Could task where a statistical regularity needs to be discovered benefit most strongly from sleep? Since the authors briefly touch on this already in the discussion (comparing their task and the NRT findings by Lacaux et al.) it might be worth introducing this additional idea already in the introduction. It may also explain why some studies did not find effects of sleep on insight (all using different tasks/problems).

2. What were the differences in paradigm between previous study and sleep study? Was there a delay time? If not, this comparison must be treated with caution because incubation in wake might similarly facilitate insight. I concur with the authors (although the group allocation was not randomized) that a comparison of participants who slept with those who did not fall asleep, in combination with their previous results, strongly suggests that sleep was key here. Please add a note of caution to the discussion/results section that this was not an experimental manipulation and strictly, causality can thus not be inferred. The correlative evidence, however, is strong and highly convincing.

3. p. 8 Could you elaborate on your nested models comparisons? In my understanding, a huge advantage of this approach is that you can actually statistically compare whether models are equivalent. Is a model with only sleep stage significantly worse than the full model with sleep stage and spectral slope? Is a model with only spectral slope significantly worse than the full model? From the way I read this paper, Fig. 3 displays nominal differences in AIC for individual electrodes? While this is informative, running statistical model comparisons would further strengthen these conclusions.

4. p.10: please qualify that the spectral slope may offer a more fine-grained measure on some aspects related to sleep depth - I believe it is still unclear in how far it may be a "better" measure, more generally, for determining how deeply a person is asleep.

5. Discussion: Please elaborate on how renormalizing excitability may translate to spectral slope and the current findings. The steep spectral slopes during sleep might reflect an active inhibition process that then leads to renormalization of excitability. The idea of synaptic homeostasis would suggest a flatter E/I slope after sleep (i.e. renormalization) than before sleep (flatter slope = higher excitation, steeper slope higher inhibition), if I reason correctly? This is a complex matter and it may be helpful for readers unfamiliar with these concepts to elaborate on the assumed mechanisms and how they translate to inhibition/excitation in neural networks.

6. Discussion: following up on the authors' thoughts on proposed mechanisms, particularly renormalization, I agree that insight might be facilitated in the N2 participants because they face the task with a "cleaner slate" after the nap. This is a different mechanism than that suggested by the original sleep and insight study, linking reactivation with gaining insight over sleep. It does fit suggestions in the insight literature that e.g. a relaxation of previously established constraints might underlie at least part of the processes leading to insight. I would really appreciate if the authors could discuss this in more detail. It addresses a new angle in sleeps role in insight and may thus encourage relevant discussion in the field.

7. Supplemental Information: It is great that the reported results hold when considering participants' classification on how deeply they slept and an automatic sleep stage classifiers categorization, but it is unclear to me, what these analyses add to the results reported in the main text. Manual sleep scoring still is the gold standard in the field and either algorithm based scoring or subjective ratings should mainly add noise to the data? Maybe I am missing an obvious point here, if yes, please consider adding a more explicit reasoning on why these additional analyses were run and what further information they provide to the paper.

8. Supplemental Info/Methods: maybe I missed it, but please report that there were no differences in vigilance (PVT) across groups in a brief statement in the main text.

9. I am not a native English speaker but I stumbled across the following phrases and wanted to draw the authors' notice to them:

- On a behavioral level � At a behavioral level, p.1

- Similarly to the NRT, Similar to the NRT p.2

- "Having an insight"? Experiencing insight? Moment of insight? P.1, p.3

Reviewer #2: The authors aimed at replicating previous findings by Lacaux et al., 2021 indicating that only N1 sleep (compared to N2 sleep and wakefulness) predicted insight. Using a perceptual insight task, the current study found greater insight likelihood for participants reaching N2 sleep compared to participants reaching only N1 sleep or staying awake during a 20-min nap opportunity. N2 sleep did not increase overall accuracy or other characteristics of insight. Explorative analyses of the aperiodic slope in the fronto-central EEG showed an association with insight, with steeper slopes being associated with a higher likelihood of insight, and adding here the sleep stage to the statistical model did not improve model fit. There was no association between oscillatory power and insight likelihood. 

This an interesting study. The link of insight to aperiodic activity is a novel finding that, in my view, could substantially advance research in this field. Overall, the methods are sound. I see only minor points that revolve around providing more methodological details for replication and conceptual clarifications.. 

- As I see, the concept of "regularization" is of central importance as it provides the rationale for analyzing aperiodic EEG activity. It should be more clearly described and defined such that readers (lacking computational expertise) understand how it relates to neuronal network activity. 

- Related to this first point, I did not understand (in the Discussion) how synaptic renormalization leads to increased regularization. Moreover, synaptic renormalization is commonly ascribed to SWS (e.g., Liu, Niethard et al. 2024, PLoSBiol). I would expect insight to be related to SWA-related parameters If synaptic renormalization is the critical factor. Overall, the discussion regard this point is not convincing and needs to be sharpened.

- Was the aperiodic slope calculated for the whole recording of an individual? Or was it for the N2 group only calculated for epochs with N2 and the N1 group only for epochs with N1 and so on? This issue is also related to the second question how much N1 sleep did the N1 group have? For the N2 group the distribution of sleep stages should be reported in more detail. For example, it might also be interesting to know if some participants reached N3. 

- The authors provide bayes factors to strengthen their results. Please provide some information on this analysis. Was it done in R, which library was used? Where the default priors used?

- For the GLMs I was wondering what kind of contrasts were used, especially for factors with 3 levels? Were these contrasts considered at all, or just the post-hoc comparisons? For the GLMs more information on the model output is desirable. Of course, AIC is helpful for model comparison, however, especially for the models reported in Fig. 3b, one just knows that one model is a better fit than the other but the information whether the better fit actually is predictive is missing. I understand that more specific information is just reported for one exemplary channel, but please provide more information than just the beta coefficient. 

- What was the rationale for modelling sleep stage and spectral slope additively. Did the authors check whether there are any interactions? Wouldn't it be possible that only in the N2 group the slope is predictive of insight?

- Please provide more information regarding the settings for the analysis using FOOOF. The frequency range for the power calculation is not necessarily the same as for the FOOOF calculation. Was a "knee" fitted? What was the maximum number of peaks to be fitted, the maximum peak height etc. 

- Analyses of aperiodic corrected power: since the authors preregistered to explore an association between delta power and insight likelihood, I was wondering why this analysis is missing in the manuscript. 

Reviewer #3: This paper is an interesting follow up to a very high profile paper from the same group which suggested that N1 sleep is highly beneficial for the type of insight involved in the number reduction task, while obtaining N2 sleep nullified these benefits. In the current study the authors have developed a new task which attempts to test the same kind of insight in a similar (but distinct) way. Thus, participants have to realise that they can use the colour of moving dots as a clue to their direction of movement. They performed this task before and after a 20 minute daytime nap opportunity in a dark room. Contrary to their prior findings, data in this study suggested that N2 sleep (rather than N1) facilitated insight. The data also seem to suggest a link between aperiodic brain activity (measured using spectral slopes) and insight. These results are surprising in the context of the prior work on N1 sleep and insight, however the literature is very small so there is a need for more studies. The question is thus interesting and topical. However I have quite a few concerns:

 1. The sleep duration seems very short and they did not provide the specific sleep parameters (e.g.: the time of wake, N1 and N2). The classification of subjects only by reaching N2 is concerning, as extremely brief periods in N2 (such as 1-2 minutes) could undermine the validity of the results.

 2. I don't know whether all the subjects did the experiment at the same time of day. Could circadian rhythms have influenced the findings?

 3. It isn't clear why the current results are so different from the prior findings regarding N1 and insight. This is discussed, but I feel it deserves a more in depth treatment.

 4. I find it surprising that insight proportions differ significantly between the groups (wake, N1, and N2), yet task performance remains unaffected. After participants learned the color rule, the task should theoretically have been easy to complete with high accuracy. I expected the N2 group to outperform the other two. Could this be due to practice effects? Despite not learning the insight rule, perhaps the participants were able to perform well after repeated trials. However, in Figure 2D, there seems to be a marked performance improvement in every group from Block 0 to Block 1.

 5. For the FOOOF analysis, they found that the slope of 1/f, rather than the oscillation, correlates to insight likelihood. Although 1/f is normally considered as a background noise, it did have some neural information itself. However, the rationale behind why the slope alone predicts results better than a model including both slope and sleep stage was not clearly explained.

 6. A minor observation: In Fig1D, the motion coherence values on the x-axis appear inconsistent with those in Fig. 1C, e.g.: the light green one is 76% in 1c but seem to be 45% in 1d.

---

## [Decision Letter · Decision Letter 2]

11 Apr 2025

Dear Dr Schuck,

Thank you for your patience while we considered your revised manuscript "N2 Sleep Inspires Insight" for publication as a Research Article at PLOS Biology. This revised version of your manuscript has been evaluated by the PLOS Biology editors, the Academic Editor and the original reviewers.

Based on the reviews, we are likely to accept this manuscript for publication, provided you satisfactorily address the remaining points raised by the reviewer 2. 

**IMPORTANT: Please also make sure to also address the following data and other policy-related requests:

1) TITLE: we would like to suggest that the title be expanded a bit to include more details about what was done here, as we think this will make it more accessible. Specifically, we suggest you change the title to: 

"N2 Sleep promotes the occurrence of 'aha' moments in a perceptual insight task"

2) ETHICS STATEMENT: Please update the ethics statement in your manuscript to indicate whether informed consent was written or verbal. Please also indicate whether the experiments were conducted according to the principles expressed in the Declaration of Helsinki. 

3) DATA/CODE: I see that your data availability statement says "Raw data (and code) will be made publicly available upon publication here (https://osf.io/z5rxg

^^Please do go ahead and make the raw data and code available at this time, as that will be needed for publication. 

Details on our data policy, which requires that all data be made available without restriction, can be found here: 

http://journals.plos.org/plosbiology/s/data-availability. For more information, please also see this editorial: http://dx.doi.org/10.1371/journal.pbio.1001797

We also require that if you have generated any custom code during the course of this investigation, please make it available without restrictions. Please ensure that the code is sufficiently well documented and reusable, and that your Data Statement in the Editorial Manager submission system accurately describes where your code can be found. The code will also need a DOI.

We expect to receive your revised manuscript within two weeks. 

*Published Peer Review History*

*Press*

Sincerely,

Luke

Lucas Smith, Ph.D.

Senior Editor

lsmith@plos.org

PLOS Biology

Reviewer remarks:

Reviewer #1: I thank the authors for their thorough responses and revisions to their paper. I have no further comments.

Reviewer #2: All points raised have been satisfactorily answered. However, now knowing that the aperiodic slope has been calculated across the whole 20-min period independent of the sleep stage, I feel that the conclusions drawn in the discussion are not entirely reflected by the results. The discussion should very clearly point out again that the aperiodic slope has been calculated across a mixture of wake and sleep epochs for the N1 and N2 groups and thus reflects sleep depth only to a limited degree. Conclusions like "Hence, the fact that the spectral slope predicts insight beyond sleep stages alone suggests that deeper sleep is needed for insight." doesn't seem to be completely supported by the current analysis. I was also wondering whether the authors might add results from an analysis of the slope for the respective stages separately (i.e. considering only N1 epochs for N1 group and N2 epochs in the N2 group).

Reviewer #3: The authors have addressed all of our comments and we now feel the manuscript is appropriate for publication.

---

## [Editor Report · Decision Letter 3]

30 Apr 2025

Dear Nico,

Thank you for the submission of your revised Short Report "N2 sleep promotes the occurrence of ’aha’ moments in a perceptual insight task" for publication in PLOS Biology and thank you for addressing the last reviewer and editorial requests in this revision. On behalf of my colleagues and the Academic Editor, Pierre-Hervé Luppi, I am pleased to say that we can in principle accept your manuscript for publication, provided you address any remaining formatting and reporting issues. These will be detailed in an email you should receive within 2-3 business days from our colleagues in the journal operations team; no action is required from you until then. Please note that we will not be able to formally accept your manuscript and schedule it for publication until you have completed any requested changes.

**IMPORTANT: as you address any formatting requests to come, please also address the following editorial requests. 

1 - As noted over email, I have updated your data availability statement to include the DOI linked to your Github deposition. Please do take a quick look to make sure everything looks OK after this change. 

2 - I noticed that there are some text included in the supplemental material. Please move that into the main manuscript. 

3 - Please note that we use the following naming convention for our supplemental figures: 'S1 Fig.', 'S2 Fig.', and so on. Please update the names of your supplemental figures accordingly. 

4 - Please add a sentence to each figure legend, including the supplemental, directing readers to the underlying data. For example, you can add the sentence "The underlying data for this figure can be found at ___" (and then cite the relevant DOI)

PRESS

We frequently collaborate with press offices. If your institution or institutions have a press office, please notify them about your upcoming paper at this point, to enable them to help maximize its impact. If the press office is planning to promote your findings, we would be grateful if they could coordinate with biologypress@plos.org. If you have previously opted in to the early version process, we ask that you notify us immediately of any press plans so that we may opt out on your behalf.

Sincerely, 

Luke 

Lucas Smith, Ph.D.

Senior Editor

PLOS Biology

lsmith@plos.org